# Improved residual fat malabsorption and growth in children with cystic fibrosis treated with a novel oral structured lipid supplement: A randomized controlled trial

**Virginia A. Stallings[1,2]\*, Alyssa M. Tindall[1], Maria R. Mascarenhas[1,2], Asim Maqbool[1,2], Joan I. Schall[1]**

**1** Division of Gastroenterology, Hepatology and Nutrition, Children's Hospital of Philadelphia, Philadelphia, PA, United States of America, **2** Department of Pediatrics, University of Pennsylvania Perelman School of Medicine, Philadelphia, PA, United States of America

\* stallingsv@email.chop.edu

## Abstract

### Background

In the primary analysis of a 12-month double-blind randomized active placebo-controlled trial, treatment of children with cystic fibrosis (CF) and pancreatic insufficiency (PI) with a readily absorbable structured lipid (Encala™, Envara Health, Wayne, PA) was safe, well-tolerated and improved dietary fat absorption (stool coefficient of fat absorption [CFA]), growth, and plasma fatty acids (FA).

### Objective

To determine if the Encala™ treatment effect varied by severity of baseline fat malabsorption.

### Methods

Subjects (n = 66, 10.5±3.0 yrs, 39% female) with baseline CFA who completed a three-month treatment with Encala™ or a calorie and macronutrient-matched placebo were included in this subgroup analysis. Subjects were categorized by median baseline CFA: low CFA (<88%) and high CFA ($\geq$88%). At baseline and 3-month evaluations, CFA (72-hour stool, weighed food record) and height (HAZ), weight (WAZ) and BMI (BMIZ) Z-scores were calculated. Fasting plasma fatty acid (FA) concentrations were also measured.

### Results

Subjects in the low CFA subgroup had significantly improved CFA (+7.5±7.2%, mean 86.3 ±6.7, p = 0.002), and reduced stool fat loss (-5.7±7.2 g/24 hours) following three months of Encala™ treatment. These subjects also had increased plasma linoleic acid (+20%), α-linolenic acid (+56%), and total FA (+20%) (p$\leq$0.005 for all) concentrations and improvements in HAZ (0.06±0.08), WAZ (0.17±0.16), and BMIZ (0.20±0.25) (p$\leq$0.002 for all). CFA and FA

**Data Availability Statement:** All relevant data are within the manuscript.

**Funding:** The NIH provided all funding for this study through a Small Business Innovation Research (SBIR) program award. This grant funded the development and production of LXS (now known as Encala™) at Avanti Polar Lipids, the SBIR awardee and supplied the LXS and placebo for the clinical trial conducted at Children's Hospital of Philadelphia. Envara Health was founded in 2018 to develop therapeutic nutrition products related to the LXS (now Encala™) technology. Envara Health provided no funding or products for this study.

**Competing interests:** The authors have declared that no competing interests exist.

were unchanged with placebo in the low CFA group, with some WAZ increases (0.14±0.24, p = 0.02). High CFA subjects (both placebo and Encala™ groups) had improvements in WAZ and some FA.

## Conclusions

Subjects with CF, PI and more severe fat malabsorption experienced greater improvements in CFA, FA and growth after three months of Encala™ treatment. Encala™ was safe, well-tolerated and efficacious in patients with CF and PI with residual fat malabsorption and improved dietary energy absorption, weight gain and FA status in this at-risk group.

## Introduction

The treatment of dietary fat malabsorption to optimize growth and nutritional status in patients with cystic fibrosis (CF) and pancreatic insufficiency (PI) is a key challenging component of standard clinical care[1]. Although CFTR modulators have shown great promise improving nutritional status[2], there is limited evidence on the effect of modulators on fat absorption. One study reported an improvement in fat absorption with Ivacaftor therapy[3], but there are no data in individuals with more severe (non-gating) mutations. Even with recommended doses of pancreatic enzyme medications, residual dietary fat malabsorption is common and contributes to weight and stature growth faltering in children and poor weight status in adults. These conditions are associated with a decline in lung function and reduced survival[4–8]. To address this clinical need, a readily absorbable structured lipid technology (Encala™ [previously LYM-X-SORB™], Envara Health, Wayne, PA) was developed and tested as an oral nutritional supplement to increase dietary fat absorption and to increase effective caloric intake. Encala™ was evaluated in a randomized placebo controlled trial[9] that reported the supplement was safe and well tolerated in children and adolescents with CF and PI, and treatment improved dietary fat absorption, growth, choline and essential fatty acid (FA) status in the participants over 12 months of treatment [10, 11]. The objective of this subgroup analysis was to determine if the effect of three months of Encala™ treatment varied by subject degree of dietary fat malabsorption at baseline.

## Methods

### Participants

Subjects ages 5.0 to 17.9 years with CF and PI and mild to moderate lung disease were recruited from ten CF Centers and evaluated between March 2007 and May 2011 (**Fig 1**). Informed, verbal assent was obtained from subjects 6.0 to <18.0 years and informed, written consent from parents/legal guardians of subjects <18 years.

### Inclusion/exclusion criteria

The complete inclusion/exclusion criteria, study design and method details, Encala™ composition and the results for Encala™ impact on choline and fatty acid status on the full study cohort have been previously reported[10–12]. Liver disease and any other significant diagnosis that might impact dietary intake, growth or body composition were exclusion criteria for subjects to participate in this study. There were no subjects with cholestatic liver disease.

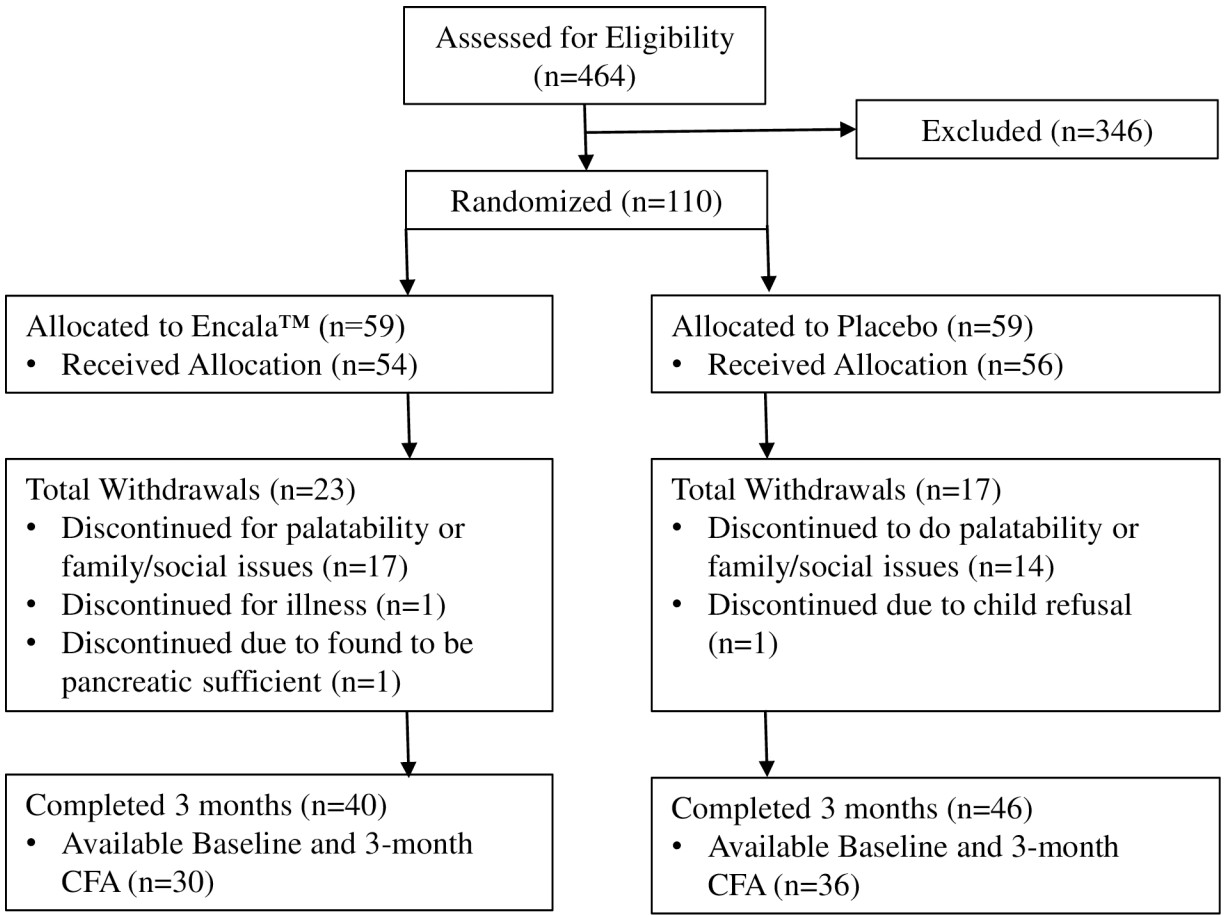

**Fig 1. Outlines the enrollment of subjects, their allocation to treatment, disposition status and how they were analyzed in this trial.**

## Design

In this double-blind placebo-controlled study, subjects were randomized in a 1:1 ratio to daily supplementation to be consumed as part of each meal and snack with either Encala™ or placebo. The random allocation sequence was generated by the CHOP research pharmacy who assigned participants to groups with stratification for age and sex; the rest of the research team was blinded to treatment allocation. Encala™ powder was mixed with a wide range of participant-selected foods and beverages and was comprised of lysophosphatidylcholine (LPC), monoglycerides and fatty acids. LPC is water-soluble, does not require lipase for digestion/absorption of associated fatty acids and generally fosters lipid absorption in the upper gastrointestinal tract [9, 13]. Encala™ was complexed to sugar and wheat flour, to provide a taste neutral, dissolvable powder. The placebo was a powder of similar appearance, taste and consistency, composed of trans-fat free vegetable shortening, flaxseed triglyceride, and sunflower triglyceride. The placebo and Encala™ had similar calories (152 kcal/32g packet), total fat (5.4g/32g packet), and macronutrient distribution (protein 6%, carbohydrate 58%, lipid 34% kcal). Subjects age 5.0–11.9 years received two packets/d (64 g powder) providing 304 kcal/d, and age subjects 12.0–17.9 years received three packets/day (96 g powder) providing 456 kcal/d. Both Encala™ and placebo were provided in sealed packages with identical appearances so participants and staff were blinded to group assignment. All subjects continued their

pancreatic enzyme regime and other aspects of care (medications, physical therapy, and diet) as prescribed by their CF center.

All study visits were conducted at Children's Hospital of Philadelphia (CHOP) at baseline, three, and 12 months and the protocol was approved by the CHOP Institutional Review Board (IRB) and each participating CF Center (Eastern Virginia Medical School IRB, Children's National Medical Center IRB, Yale University School of Medicine IRB, University of Virginia IRB, Schneider Children's Hospital IRB (now: Cohen Children's Medical Center), St. Joseph's Children's Hospital IRB, and Monmouth Medical Center IRB). The data from the baseline and three-month visits were selected for this analysis. This protocol was registered as Study of Lym-X-Sorb (Encala™) to Improve Fatty Acid and Choline Status in Children with Cystic Fibrosis and Pancreatic Insufficiency, NCT00406536.

Pulmonary function was assessed and predicted percentage $FEV_1$ calculated[14, 15]. Body mass index (BMI) was calculated ($kg/m^2$) from weight using a digital scale (Scaletronix, White Plains, NY) and standing height using a stadiometer (Holtain, Crymych, UK) measured by research staff. Weight, height and BMI were compared to Centers for Disease Control reference standards to generate age- and sex-specific Z scores[16]. Total body fat mass, lean body mass, and percentage fat were measured by whole-body dual-energy X-ray absorptiometry (DXA; Delphi A, Hologic, Inc., Bedford, MA).

Dietary intake was measured using 3-day weighed records and analyzed for energy and fat intake (Nutrition Data System, Minneapolis, MN)(17). Subjects were trained for diet collection by research staff and provided food scales, measuring cups and spoons. Energy intake was reported as kcal/day and as percent Estimated Energy Requirement (%EER) for active children and adolescents[17, 18]. A 72-hour stool sample was collected and total fat content determined (Mayo Medical Laboratories, Rochester, MN), and CFA was calculated[19]. For the 3-month dietary assessment, the calorie and fat content of Encala™ and placebo supplements were included in the daily energy and fat intake data, adjusted for adherence[11].

Quantitation of morning fasting plasma FA was performed in two steps: 1) acid-base hydrolysis; 2) hexane extraction/derivatization with pentafluorobenzyl bromide (Mayo Medical Laboratories). Separation/detection was accomplished by capillary gas chromatography electron-capture negative ion-mass spectrometry, with quantitation based on analysis in selected ion-monitoring mode using stable isotope-labeled internal standards[20].

## Statistics

Descriptive statistics are presented as frequency counts and percentages for categorical variables and mean ± standard deviation (SD) for continuous variables. In addition to the randomization groups (placebo vs. Encala™) for this analysis, subjects were assigned to one of two fat malabsorption subgroups based upon their CFA at baseline: 1) low baseline CFA subgroup with CFA below the cohort median of 88%; and 2) high baseline CFA subgroup with CFA equal to or above the median of 88%. Two-sample t-test (unpaired) or Mann-Whitney U test for continuous variables and chi-square tests of independence for categorical variables compared characteristics at baseline between randomization groups (placebo vs. Encala™), and also between low and high baseline CFA subgroups. Two-sample t-tests were also used to determine differences between randomization groups for outcomes at three months and for the 3-month change in outcomes. Paired t-tests were used to determine significance of change in outcomes within randomization groups for the low CFA and high CFA subgroups separately. Stata 12.0 (Stata Corporation, College Station, TX) was used with significance at 0.05.

## Results

Eighty-six participants completed three months treatment on either Encala™ or placebo, and this subgroup analysis included the 66 children and adolescents (10.5±3.0 yrs, 39% female) who had both baseline CFA and 3-month visit assessments were the basis of this analysis. Table 1 provides the characteristics of the sample at baseline for all 66 subjects and for those in the high and low baseline CFA subgroups. At baseline, all subjects had mild to moderate lung disease ($FEV_1$ 99±22% predicted), and suboptimal growth status as indicated by Z scores for weight and height. Subjects in the low baseline CFA subgroup were significantly older, more likely to be homozygous for the F508del allele, and had lower BMI Z scores and %EER energy intake. Plasma fatty acid status, $FEV_1$ and daily dietary intake of fat and energy were similar between CFA subgroups at baseline.

Table 1 also provides characteristics of the sample at baseline by randomization groups (placebo vs. Encala™). Subjects randomized to Encala™ had significantly lower BMI Z scores than those randomized to placebo (-0.39±0.66 vs. 0.0±0.68, p<0.05), and were otherwise similar in growth, body composition, dietary intake and plasma fatty acid status. A greater proportion of the Encala™ group were in the low baseline CFA subgroup than the placebo group (63 vs. 39%, p<0.05), although there was no significant difference in mean CFA or stool fat loss at baseline.

Table 2 presents the change in outcomes from baseline to three-months for subjects receiving placebo or Encala™ within each of the baseline CFA subgroups. For those in the low CFA subgroup, three-month Encala™ treatment improved CFA significantly (7.5±7.2%, 78.9±7.5 to 86.3±6.7%, p = 0.002), with a significant drop in stool fat loss (-5.7±7.2 g/24 hours) and no change in dietary fat intake. At the same time, plasma linoleic acid increased 20%, α-linolenic acid 56%, γ-linolenic 51%, docosapentaenoic acid 45%, and arachidonic acid by 23% (p<0.05). Total plasma fatty acid concentration increased by 20% (p≤0.005), with similar increases in total saturated, monounsaturated and polyunsaturated fatty acids, and HAZ (0.06 ±0.08), WAZ (0.17±0.16), and BMIZ (0.20±0.25) all increased (p≤0.002). CFA status did not change with placebo treatment in the low baseline CFA subgroup. The low baseline CFA group receiving placebo treatment had significant reductions in dietary monounsaturated fatty acids (-0.3 ± 0.4 mmol/L) and plasma docosahexaenoic acid (-17%) with an increase in plasma eicosapentaenoic acid (31%) and WAZ (0.12±0.17, p's<0.05). For subjects in the high baseline CFA subgroup, CFA% (-1.2±4.8% on placebo and -4.4±6.3% on Encala™), stool fat (0.3±4.5 g/d on placebo and 2.9±7.3 g/d on Encala™) and dietary fat intake did not change with either placebo or Encala™ treatment. WAZ improved in both groups (0.13±0.28 for placebo and 0.14±0.21 for Encala™, p<0.05).

## Discussion

Encala™ was evaluated initially as a nutritional supplement in 2002 in children with CF and PI [9]. This novel compound was synthesized from known lipolysis products of dietary fats, including lysophosphatidycholine, monoglyceride and fatty acids. Lepage, et al.[9] published the results of two studies testing the safety and efficacy of Encala™. In the preliminary study, a liquid Encala™ test meal was consumed by healthy adults and by adolescents with CF and PI taking no pancreatic enzyme medication and plasma triglyceride absorption followed over 12 hours. Total triglycerides absorbed from Encala™ did not differ between the CF and healthy subjects and established that pancreatic enzymes were not required for Encala™ digestion and absorption in the context of CF and PI. In the 12-month double blind randomized, placebo-controlled CF clinical trial that followed, Encala™ and an active, isocaloric placebo supplements were tested in a cookie form in children and adolescents. The results showed that the Encala™

**Table 1. Characteristics at baseline for CFA subgroups and for randomization groups.**

| | | By CFA Subgroup | | By Randomization Group | |
|---|---|---|---|---|---|
| | **All** | **High CFA (≥88%)** | **Low CFA (<88%)** | **Placebo** | **Encala™** |
| Characteristic | n = 66 | n = 33 | n = 33 | n = 36 | n = 30 |
| Sex, % males | 61 | 55 | 67 | 67 | 53 |
| F508del homozygous, % | 55 | 42 | **67***  | 50 | 60 |
| Age, yr | 10.5 ± 3.0 | 9.6 ± 2.7 | **11.5 ± 2.9*** | 10.4 ± 3.1 | 10.7 ± 2.9 |
| FEV$_1$% [a] | 99 ± 22 | 101 ± 24 | 98 ± 20 | 104 ± 22 | 94 ± 20 |
| **Growth and Body Composition** | | | | | |
| Height for age Z score | -0.52 ± 0.93 | -0.54 ± 1.13 | -0.49 ± 0.69 | -0.57 ± 0.90 | -0.46 ± 0.97 |
| Weight for age Z score | -0.45 ± 0.72 | -0.34 ± 0.88 | -0.55 ± 0.52 | -0.35 ± 0.77 | -0.57 ± 0.66 |
| BMI for age Z score | -0.17 ± 0.69 | 0.02 ± 0.73 | **-0.37 ± 0.60*** | 0.00 ± 0.68 | **-0.39 ± 0.66*** |
| Whole Body DXA | | | | | |
| FFM, kg | 26.3 ± 9.7 | 23.9 ± 8.2 | 28.7 ± 10.6* | 26.7 ± 10.9 | 25.8 ± 8.2 |
| FM, kg | 7.1 ± 2.8 | 6.9 ± 2.8 | 7.2 ± 2.8 | 6.9 ± 2.5 | 7.3 ± 3.1 |
| Fat, % | 21.6 ± 5.8 | 22.7 ± 5.2 | 20.6 ± 6.3 | 21.4 ± 6.3 | 21.9 ± 5.3 |
| **Coefficient of Fat Absorption** | | | | | |
| Stool fat, g/day | 16.4 ± 13.1 | 8.3 ± 4.8 | **24.4 ± 13.8***  | 15.6 ± 14.6 | 17.3 ± 11.1 |
| Dietary fat intake, g/day | 103.3 ± 35.4 | 108.5 ± 41.4 | 98.0 ± 28.0 | 105.9 ± 38.7 | 100.2 ± 31.4 |
| CFA, % | 83.9 ± 11.2 | 92.4 ± 2.6 | **75.3 ± 9.9***  | 85.1 ± 12.1 | 82.4 ± 10.2 |
| CFA < Median (88%), % | 50 | 0 | 100 | 39 | **63*** |
| Dietary energy intake, Kcal/day | 2556 ± 692 | 2611 ± 792 | 2501 ± 582 | 2588 ± 740 | 2518 ± 639 |
| EER % | 124 ± 29 | 132 ± 37 | **116 ± 17*** | 124 ± 31 | 123 ± 28 |
| **Plasma Fatty Acids, nmol/ L [b]** | | | | | |
| Linoleic acid | 2292 ± 514 | 2348 ± 428 | 2239 ± 584 | 2308 ± 562 | 2271 ± 453 |
| α-Linolenic acid | 51.4 ± 24.4 | 52.0 ± 21.0 | 50.9 ± 27.6 | 52.6 ± 28.7 | 49.9 ± 18.0 |
| γ-Linolenic acid | 65.4 ± 28.7 | 61.4 ± 28.7 | 69.2 ± 28.6 | 63.7 ± 29.4 | 67.6 ± 28.1 |
| Arachidonic acid | 438 ± 153 | 445 ± 167 | 432 ± 141 | 441 ± 160 | 436 ± 145 |
| Eicosapentaenoic acid | 37.1 ± 23.4 | 37.3 ± 23.1 | 36.8 ± 24.1 | 36.7 ± 21.3 | 37.5 ±26.4 |
| Docosahexaenoic acid | 61.1 ± 23.6 | 62.3 ± 23.6 | 60.0 ± 28.8 | 62.1 ± 24.8 | 59.9 ± 28.4 |
| Docosapentaenoic acid | 36.2 ± 16.0 | 37.5 ±17.8 | 35.0 ± 14.3 | 37.5 ± 17.5 | 34.5 ± 14.1 |
| **Plasma Fatty Acid Group, mmol/L [b]** | | | | | |
| Saturated | 3.2 ± 0.7 | 3.2 ± 0.7 | 3.2 ± 0.7 | 3.2 ± 0.7 | 3.1 ± 0.7 |
| Monounsaturated | 2.4 ± 0.5 | 2.3 ± 0.5 | 2.4 ± 0.7 | 2.4 ± 0.7 | 2.3 ± 0.5 |
| Polyunsaturated | 3.2 ± 0.7 | 3.1 ± 0.6 | 3.2 ± 0.8 | 3.2 ± 0.8 | 3.1 ± 0.6 |
| Total Fatty Acids | 8.8 ± 1.8 | 8.7 ± 1.7 | 8.8 ± 2.0 | 8.8 ± 2.0 | 8.7 ± 1.7 |

CFA: coefficient of fat absorption; DXA: dual x-ray absorptiometry; FFM: fat free mass; FM: fat mass; Fat: fat mass as percent total body mass; FEV$_1$%: Forced expiratory volume at 1 second percent predicted value.

Data are presented as mean ± SD and frequency (percentage) for categorical data.

Bolded values show statistical significance.

*Significantly different between randomization groups by unpaired t test for continuous variables and χ-squared test or Fisher's exact test for categorical variables, p<0.05,

***p<0.001

[a] All, n = 63; High CFA, n = 32; Low CFA, n = 31: Placebo, n = 36; Encala™, n = 27

[b] All, n = 64; High CFA, n = 31; Low CFA, n = 33: Placebo, n = 36; Encala™, n = 28

supplement was safe and was a well-absorbed source of dietary fat energy that Encala™ use led to better clinical outcomes in growth, fat-soluble vitamins and essential fatty acid status in this cohort of children with CF and PI. However, the cookie format and taste were not optimal

**Table 2. Outcomes by coefficient of fat absorption subgroups: Encala™ vs. placebo.**

| | | CFA < Median (<88%) | | | | | CFA ≥ Median (≥88%) | | | |
|---|---|---|---|---|---|---|---|---|---|---|
| Characteristic | n | Baseline | 3 Months | 3-Month Change | P | n | Baseline | 3 Months | 3-Month Change | P |
| **CFA, %** | | | | | | | | | | |
| Placebo | 12 | 72.5 ± 11.8 | 71.1 ± 26.2 | -1.4 ± 28.5 | ns | 18 | 92.8 ± 2.4 | 91.1 ± 5.9 | -1.2 ± 4.8 | ns |
| Encala™ | 14 | 78.9 ± 7.5 | 86.3 ± 6.7 ϯ | **7.5 ± 7.2** | 0.002 | 8 | 91.4 ± 2.9 | 87.1 ± 8.3 | -4.4 ± 6.3 | ns |
| **Stool Fat, g/day** | | | | | | | | | | |
| Placebo | 12 | 30.4 ± 16.6 | 30.1 ± 25.3 | -0.3 ± 32.1 | ns | 18 | 7.6 ± 4.2 | 7.9 ± 5.2 | 0.3 ± 4.5 | ns |
| LXS | 14 | 19.9 ± 10.7 | 14.3 ± 90.4 ϯ | **-5.7 ± 7.2** | 0.012 | 8 | 9.6 ± 4.7 | 12.6 ± 11.1 | 2.9 ± 7.3 | ns |
| **Dietary Total Fat Intake, g/day** | | | | | | | | | | |
| Placebo | 12 | 111.6 ± 28.1 | 109.2 ± 22.6 | -2.4 ± 41.9 | ns | 18 | 108.4 ± 46.6 | 100.6 ± 33.6 | -7.8 ± 35.5 | ns |
| Encala™ | 14 | 92.5 ± 25.8 | 106.5 ± 33.8 | 14.0 ± 30.6 | ns | 8 | 109.3 ± 32.8 | 91.7 ± 38.3 | -17.6 ± 36.1 | ns |
| **Dietary Saturated FAs, g/day** | | | | | | | | | | |
| Placebo | 12 | 41.0 ± 10.5 | 41.3 ± 11.5 | 0.3 ± 15.9 | ns | 18 | 36.9 ± 9.7 | 34.2 ± 11.3 | -2.7 ± 10.0 | ns |
| Encala™ | 14 | 35.6 ± 12.6 | 37.6 ± 13.3 | 2.1 ± 10.7 | ns | 8 | 40.7 ± 12.6 | 30.8 ± 16.5 | -9.8 ± 13.0 | ns |
| **Dietary Monounsaturated FA, g/day** | | | | | | | | | | |
| Placebo | 12 | 40.2 ± 10.8 | 36.4 ± 8.0 | -3.8 ± 16.0 | ns | 18 | 40.3 ± 22.7 | 34.3 ± 14.9 | -6.0 ± 16.4 | ns |
| Encala™ | 14 | 29.6 ± 6.9 | 34.1 ± 10.6 | 4.4 ± 8.9 | ns | 8 | 38.5 ± 12.0 | 32.1 ± 14.2 | -6.4 ± 15.4 | ns |
| **Dietary Polyunsaturated FAs, g/day** | | | | | | | | | | |
| Placebo | 12 | 20.8 ± 7.8 | 22.8 ± 5.8 | 2.0 ± 8.9 | ns | 18 | 23.0 ± 18.1 | 25.2 ± 11.8 | 2.2 ± 12.7 | ns |
| Encala™ | 14 | 19.5 ± 7.3 | 26.9 ± 12.3 | **7.4 ± 12.0** | 0.039 | 8 | 21.7 ± 9.4 | 22.3 ± 8.2 | 0.5 ± 7.1 | ns |
| **HAZ** | | | | | | | | | | |
| Placebo | 14 | -0.51 ± 0.75 | -0.50 ± 0.75 | 0.02 ± 0.08 | ns | 22 | -0.60 ± 1.00 | -0.50 ± 1.03 | **0.11 ± .20** | 0.026 |
| Encala™ | 19 | -0.48 ± 0.66 | -0.41 ± 0.67 | **0.06 ± 0.08** | 0.002 | 11 | -0.43 ± 1.40 | -0.40 ± 0.41 | 0.03 ± .12 | ns |
| **WAZ** | | | | | | | | | | |
| Placebo | 14 | -0.43 ± 0.62 | -0.32 ± 0.59 | **0.12 ± 0.17** | 0.023 | 22 | -0.30 ± 0.86 | -0.17 ± 0.93 | **0.13 ± .028** | 0.038 |
| Encala™ | 19 | -0.64 ± 0.42 | -0.47 ± 0.43 | **0.17 ± 0.16** | <0.001 | 11 | -0.43 ± 0.96 | -0.29 ± 1.02 | **0.14 ± 0.21** | 0.046 |
| **BMIZ** | | | | | | | | | | |
| Placebo | 14 | -0.17 ± 0.62 | -0.04 ± 0.60 | 0.14 ± 0.24 | ns | 22 | 0.12 ± 0.71 | 0.19 ± 0.81 | 0.07 ± 0.38 | ns |
| Encala™ | 19 | -0.51 ± 0.56 | -0.31 ± .51 | **0.20 ± 0.24** | 0.002 | 11 | -0.18 ± 0.78 | -0.00 ± 0.79 | 0.18 ± 0.38 | ns |
| **Linoleic Acid, nmol/L** | | | | | | | | | | |
| Placebo | 14 | 2309 ± 778 | 2159 ± 704 | -150 ± 686 | ns | 22 | 2306 ± 390 | 2480 ± 513 | **173 ± 382** | 0.046 |
| Encala™ | 19 | 2187 ± 404 | 2625 ± 695 | **438 ± 511** ϯϯ | 0.002 | 9 | 2449 ± 522 | 2574 ± 653 | 126 ± 651 | ns |
| **α-Linolenic Acid, nmol/L** | | | | | | | | | | |
| Placebo | 14 | 57.6 ± 36.4 | 57.9 ± 41.8 | -0.2 ± 44.7 | ns | 22 | 49.3 ± 22.9 | 62.8 ± 27.8 | **13.5 ± 24.4** | 0.017 |
| Encala™ | 19 | 45.9 ± 18.5 | 71.5 ± 40.3 | **25.6 ± 31.1** | 0.002 | 9 | 58.4 ± 14.3 | 79.7 ± 33.2 | 21.2 ± 36.2 | ns |
| **γ-Linolenic Acid, nmol/L** | | | | | | | | | | |
| Placebo | 14 | 74.3 ± 29.8 | 77.4 ± 40.6 | 3.1 ± 32.8 | ns | 22 | 57.0 ± 27.7 | 67.1 ± 39.5 | 10.2 ± 38.2 | ns |
| Encala™ | 19 | 65.4 ± 27.8 | 98.9 ± 69.2 | **33.5 ± 65.8** | 0.039 | 9 | 72.1 ± 29.7 | 92.8 ± 43.8 | 20.7 ± 39.4 | ns |
| **Arachidonic Acid, nmol/L** | | | | | | | | | | |
| Placebo | 14 | 457 ± 172 | 412 ± 157 | -45 ± 133 | ns | 22 | 430 ± 156 | 417 ± 125 | -13 ± 90 | ns |
| Encala™ | 19 | 413 ± 114 | 509 ± 179 | **96 ± 128** ϯϯ | 0.004 | 9 | 483 ± 196 | 470 ± 149 | -13 ± 161 | ns |
| **Eicosapentaenoic acid, nmol/L** | | | | | | | | | | |
| Placebo | 14 | 35.6 ± 11.4 | 46.5 ± 22.6 | **10.9 ± 18.2** | 0.043 | 22 | 37.5 ± 25.9 | 46.1 ± 33.4 | 8.6 ± 27.0 | ns |
| Encala™ | 19 | 37.8 ± 30.6 | 42.5 ± 18.8 | 4.7 ± 27.6 | ns | 9 | 36.9 ± 15.3 | 47.1 ± 14.1 | 10.2 ± 18.8 | ns |
| **Docosahexaenoic acid, nmol/L** | | | | | | | | | | |
| Placebo | 14 | 60.9 ± 25.2 | 50.5 ± 15.0 | **-10.4 ± 17.1** | 0.04 | 22 | 62.8 ± 25.1 | 64.8 ± 25.3 | 2.0 ± 15.9 | ns |
| Encala™ | 19 | 59.3 ± 31.8 | 67.2 ± 46.9 | 7.9 ± 25.4 ϯ | ns | 9 | 61.1 ± 21.1 | 70.2 ± 30.4 | 9.1 ± 36.5 | ns |
| **Docosapentaenoic acid, nmol/L** | | | | | | | | | | |

*(Continued)*

**Table 2.** (Continued)

| Characteristic | n | CFA < Median (<88%) | | | | n | CFA ≥ Median (≥88%) | | | |
|---|---|---|---|---|---|---|---|---|---|---|
| | | **Baseline** | **3 Months** | **3-Month Change** | **P** | | **Baseline** | **3 Months** | **3-Month Change** | **P** |
| Placebo | 14 | 38.2 ± 16.0 | 38.5 ± 16.3 | 0.3 ± 16.4 | ns | 22 | 37.0 ± 18.7 | 42.4 ± 21.2 | 5.3 ± 13.9 | ns |
| Encala™ | 19 | 32.7 ± 12.9 | 47.4 ± 22.1 | **14.6 ± 17.8 Ⱡ** | 0.002 | 9 | 38.4 ± 16.4 | 43.9 ± 15.4 | 5.4 ± 18.0 | ns |
| **Saturated FAs, mmol/L** | | | | | | | | | | |
| Placebo | 14 | 3.4 ± 0.7 | 3.3 ± 1.1 | -0.2 ± 0.6 | ns | 22 | 3.1 ± 0.7 | 3.4 ± 1.0 | 0.3 ± 0.9 | ns |
| Encala™ | 19 | 3.1 ± 0.6 | 3.7 ± 1.1 | **0.6 ± 0.9 Ⱡ** | 0.011 | 9 | 3.3 ± 0.7 | 3.8 ± 1.4 | 0.5 ± 1.1 | ns |
| **Monounsaturated FAs, mmol/L** | | | | | | | | | | |
| Placebo | 14 | 2.6 ± 0.6 | 2.3 ± 0.6 | **-0.3 ± 0.4** | 0.021 | 22 | 2.3 ± 0.7 | 2.4 ± 0.8 | 0.1 ± 0.9 | ns |
| Encala™ | 19 | 2.3 ± 0.5 | 2.8 ± 0.9 | **0.5 ± 0.7 Ⱡⱡ** | 0.012 | 9 | 2.4 ± 0.5 | 2.7 ± 0.9 | 0.3 ± 0.7 | ns |
| **Polyunsaturated FAs, mmol/L** | | | | | | | | | | |
| Placebo | 14 | 3.2 ± 1.1 | 3.1 ± 1.0 | 0.2 ± 1.0 | ns | 22 | 3.2 ± 0.7 | 3.4 ± 0.8 | 0.2 ± 0.6 | ns |
| Encala™ | 19 | 3.0 ± 0.6 | 3.7 ± 1.1 | **0.7 ± 0.8 Ⱡⱡ** | 0.001 | 9 | 3.4 ± 0.8 | 3.6 ± 0.9 | 0.2 ± 0.9 | ns |
| **Total FAs, mmol/L** | | | | | | | | | | |
| Placebo | 14 | 9.3 ± 8.1 | 8.6 ± 2.5 | -0.6 ± 1.7 | ns | 22 | 8.6 ± 2.0 | 9.2 ± 2.4 | 0.6 ± 2.2 | ns |
| Encala™ | 19 | 8.4 ± 1.6 | 10.2 ± 2.9 | **1.7 ± 2.4 Ⱡⱡ** | 0.005 | 9 | 9.1 ± 1.8 | 10.2 ± 2.9 | 1.0 ± 2.5 | ns |

CFA: Coefficient of fat absorption; HAZ: height for age Z score; WAZ: weight for age Z score; BMIZ: BMI for age score; FA: fatty acid.

Data are presented as mean ± SD for normally distributed data.

P value is testing for Time effect within treatment group using student's paired t test.

Bolded values show statistical significance.

Ⱡ Placebo and Encala™ groups different by student's unpaired t test p<0.05,

Ⱡⱡ p<0.01.

(Dr. Yesair, BioMolecular Products, personal communication), and research and development continued to address the supplement taste and format characteristics. The improved Encala™, now a taste-neutral powder that was mixed with subject-selected, preferred foods and beverages, was used in the double blind, randomized, active placebo-controlled clinical trial reported here.

The main outcomes from our RCT study showed that in a large cohort of children and adolescents with CF and PI cared for at ten different centers and on recommended CF diet and pancreatic enzyme regimens, daily oral Encala™ powder was safe and more effective than the energy and macronutrient-matched placebo. Increased dietary fat absorption (CFA) and fasting plasma fatty acid status and growth were observed[11]. Previous reports also documented that Encala™ supplementation, compared with active placebo, improved choline (serum and muscle stores), amino acid, vitamin A and E, total body muscle stores and resting energy expenditure status in these school aged children with CF, PI and mild lung disease[10, 12, 21].

In this current subgroup analysis of subjects with baseline and three-month observations, Encala™ was particularly effective in subjects who experienced more severe dietary fat malabsorption prior to intervention. Low baseline CFA value indicated more dietary fat excreted in stool and a more severe malabsorptive clinical status. In this group, Encala™ treatment improved CFA by 7.5%, a magnitude that was both highly clinically and statistically significant and improved plasma fatty acid and growth status compared to placebo. Given the success of CFTR modulators in improving nutritional status, such as BMI, in patients with some genotypes more data are also needed to determine the efficacy of nutritional treatment in patients with CF taking CFTR modulator therapies. There are limited data on the impact on fat absorption with modulator therapy, particularly in patients with more severe mutations.

Encala™ was developed to improve dietary fat absorption, especially absorption long chain fatty acids including the two essential fatty acids, and to supply more calories per gram ingested than medium chain fatty acids, which are a common ingredient in therapeutic nutrition products. Investigators have documented persistent, residual dietary fat malabsorption in spite of optimized, modern pancreatic enzyme medication and treatment regimens for patients with CF and PI for the past two decades[4, 6]. Kalivianakis et al.[6] in CF and PI studies, suggested that this residual fat loss was due to reduced gut mucosal uptake of long chain fatty acids and/or to incomplete solubilization of long chain fatty acids in the gut lumen. As such, this residual malabsorption was not the result of impaired lipolysis (pancreatic lipase mechanism), and thus will not be reduced or eliminated by increased dose of pancreatic enzyme medication[6]. Their work in a fat malabsorption rat model showed that some of the long chain fatty acid absorption impairment was attributable to chronic bile acid deficiency. Chronic bile acid depletion is an established component of the spectrum of disease in patients with CF and PI[7].

Lysophosphatidylcholine (LPC) is the backbone of the Encala™ technology and has unique metabolic characteristics; most importantly LPC directly enhances dietary fatty acid absorption in the gut, enhances transfer of fatty acids to the lymphatic system and improves fatty acid retention in mucosa[22]. Intraluminal micellar LPC is extensively absorbed in the gut[23]. In contrast, phosphatidylcholine, the common phospholipid in the diet, directly suppresses dietary fatty acid absorption as described by Homan and Hamelehle[23] in the Caco-2 cell line. Peretti et al.[24] provided a full review of the mechanisms of lipid absorption in CF.

Few patients with CF and PI have a CFA $\geq$ 93%, the generally accepted range of absorption in health children and adults[3, 19]. Encala™ supplementation addresses this residual long chain fatty acid malabsorption, as the Encala™ -delivered fatty acids are not dependent upon the action of lipase or bile acids for effective digestion and absorption in the CF gut. Despite having similar daily dietary energy and total fat intake as subjects receiving placebo, those receiving Encala™ had significantly improved absorption of linoleic, α-linolenic, γ-linolenic, arachidonic, docosapentaenoic acid and total fatty acids including saturated, monounsaturated and polyunsaturated fat classes. The improvement in fatty acid status may have also affected growth. There was improvement in growth and fatty acid status with Encala™ treatment in subjects with high baseline CFA fat malabsorption, but this improvement was less than in the low baseline CFA subgroup. The placebo group showed some improvements, as this group also received additional energy from the active placebo. This analysis demonstrates Encala™ provided greater benefits to children with CF and PI with more severe baseline residual dietary fat malabsorption. This is particularly important as these were the patients with the greatest need for nutritional treatment as they had the poorer growth and more severe disease.

Malnutrition due to chronic malabsorption is an important intervention target in CF care as it affects growth, development, pulmonary function, immune system function, and survival [25]. Participants had dietary patterns that adhered to CF recommendations, with high calorie intake (124% EER) and high fat intake (42% energy; 103 g per day)[26–28]. Participants also followed the recommended pancreatic enzyme medication use regimen for meals and snacks [29] but had a mean baseline CFA of 84%, a value lower than the 93% reference for healthy people[6, 30]. Numerous studies have demonstrated pancreatic enzyme replacement therapy (PERT) decreases fat malabsorption[26], however, PERT does not fully remedy fat malabsorption in patients with CF and PI. Further, individuals at higher risk for fat malabsorption may not attain the same improvements with enzyme treatment as those with a lower risk[31]. Woestenenk and colleagues[31] completed a retrospective cross-sectional analysis of 224 children and adolescents with CF and PI prescribed PERT. The authors reported the median CFA varied (86–91%), and 24% had a CFA below 85% and CFA did improve following PERT. The

lower CFA group also had significantly lower WAZ and BMIZ compared to the higher CFA group. Often, gastric acid-reducing agents are prescribed as adjunct therapy based on the theory that these medications reduce the gastric and duodenal acid environment and improve the efficacy of PERT. However, a recent Cochrane Database Systematic Review determined there was limited evidence that adjunct therapy to reduce gastric acid improved nutrition status[32]. New therapies, such as Encala™, are needed for patients that have residual fat malabsorption after optimizing PERT. Encala™ supplementation significantly improved CFA (+7.5%) when assessed after three months of treatment in the absence of change in dietary fat intake. The increase in CFA attained in the lower CFA group supplemented with Encala™ is clinically significant and warrants use as an adjunct therapy for individuals with CF on enzyme medications, with residual fat malabsorption.

Currently, enzyme medications are the only available treatment for dietary fat malabsorption and are a cornerstone of CF therapy. Enzyme products are a combination of all digestive enzymes and other components derived from porcine pancreas. There are non-capsule enzyme options to treat fat malabsorption, such as an immobilized lipase-only containing device available for use with enteral tube feeding (RELiZORB® Alcresta Therapeutics, Inc, Newton, Massachusetts, USA)[33], a bacterial-derived lipase-only product suspended in a liquid solution for use with meals (Burlulipase, Nordmark Arzneimittel, Uetersen, DE)[34] and a non-enteric coated enzyme powder (Viokase, Axcan Pharma US, Inc., Mont-saint-hilaire, Quebec, CA). Although lipase containing drugs improve triglyceride digestion in lipase-limited diseases, Encala™ is the only non-enzyme approach and has demonstrated improved fat absorption, growth, and nutritional status when used in conjunction with PERT. In the present study, we observed significant improvement in plasma fatty acid status following Encala™ treatment in individuals with greater fat malabsorption. Notably, supplementation increased essential fatty acid (linoleic and α-linolenic acid), arachidonic acid, docosapentaenoic acid and total fatty acid absorption. The increased dietary fat absorption and improved energy intake was reflected in the clinically significant growth in weight and height (HAZ, WAZ and BMIZ). Only WAZ improved with placebo supplementation in the lower CFA group, and this is likely due to the caloric density design of the placebo.

Given the exploratory nature of this analysis, there are some limitations. The lower and higher CFA groups in this study were defined for this study as the median CFA at baseline (88%) to provide a balanced distribution of participants. There is no generally accepted or proposed evidence-based definition for classification of individuals with CF and PI at high and low degree of fat malabsorption by CFA value. CFA is used mostly in the research setting and for FDA evaluation of new pancreatic enzyme products. For the cohort of subjects with a CFA both at baseline and the three month follow-up, there was a greater proportion of participants randomized to Encala™ in the lower CFA group. The unbalanced distribution was related to the original Encala™/placebo randomization procedure for all subjects with sex/age blocks that did not include CFA, as CFA was an outcome variable. There was no difference in mean CFA between subgroups at baseline and there were clinically significant improvements in CFA in the low CFA Encala™ group. There was variation in individual response as indicated by the standard deviation. The original study did not specify this subgroup analysis, however this additional evidence provided data and experience to inform future research and clinical care.

In summary, subjects with low baseline CFA and more severe fat malabsorption had a dramatic improvement in CFA with Encala™ treatment, accompanied by improved plasma fatty acid and growth status. Based on these results and clinical trial experience, Encala™ was well-accepted and reduced residual fat malabsorption in patients with CF and PI. These data indicate that Encala™ was efficacious in patients with CF and suggest efficacy in other exocrine pancreatic insufficiency diagnoses and malnutrition in need of improved energy absorption

and weight gain. Further investigation is needed to evaluate the effectiveness of Encala™ supplementation to improve fat absorption and treat or prevent malnutrition in patients with other diagnoses.

## Supporting information

**S1 File.**
(PDF)

**S1 Checklist.**
(DOC)

## Acknowledgments

The content is solely the responsibility of the authors and does not necessarily represent the official views of the NIH. The study was registered as: Study of Lym-X-Sorb (Encala™) to Improve Fatty Acid and Choline Status in Children with Cystic Fibrosis and Pancreatic Insufficiency, NCT00406536. https://clinicaltrials.gov/ct2/show/NCT00406536.

The authors thank the subjects, parents, other care providers, and all of the CF Centers that participated in the study: Children's National Medical Center, Washington, DC; Children's Hospital of Philadelphia, Philadelphia, PA; Monmouth Medical Center, Long Branch, NJ; The Pediatric Lung Center, Fairfax, VA; Cystic Fibrosis Center of University of Virginia, Charlottesville, VA; Children's Hospital of the King's Daughters, Eastern Virginia Medical School, Norfolk, VA; Yale University School of Medicine, New Haven, CT; Cohen Children's Medical Center, New Hyde Park, NY; St Joseph's Children's Hospital, Paterson, NJ, and the Pediatric Specialty Center at Lehigh Valley Hospital, Bethlehem, PA.. The authors also thank Norma Latham, MS, for study coordination, and Megan Johnson, Thananya Wooden, Elizabeth Matarrese, and Nimanee Harris, the staff of the CTRC at CHOP for their valuable contributions to the study.

## Author Contributions

**Conceptualization:** Virginia A. Stallings, Joan I. Schall.

**Formal analysis:** Virginia A. Stallings, Joan I. Schall.

**Funding acquisition:** Virginia A. Stallings.

**Investigation:** Virginia A. Stallings, Maria R. Mascarenhas, Asim Maqbool, Joan I. Schall.

**Methodology:** Virginia A. Stallings, Maria R. Mascarenhas, Joan I. Schall.

**Project administration:** Virginia A. Stallings.

**Supervision:** Virginia A. Stallings.

**Writing – original draft:** Virginia A. Stallings, Alyssa M. Tindall, Joan I. Schall.

**Writing – review & editing:** Virginia A. Stallings, Alyssa M. Tindall, Maria R. Mascarenhas, Asim Maqbool, Joan I. Schall.

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
