## [Decision Letter · Decision Letter 0]

26 Feb 2020

PONE-D-19-35347

Improved Residual Fat Malabsorption and Growth in Children with Cystic Fibrosis Treated with a Novel Oral Structured Lipid Supplement: A Randomized Controlled Trial

PLOS ONE

Dear Dr. Tindall,

Thank you for submitting your manuscript to PLOS ONE. After careful consideration, we feel that it has merit but does not fully meet PLOS ONE’s publication criteria as it currently stands. Therefore, we invite you to submit a revised version of the manuscript that addresses the points raised during the review process.

We would appreciate receiving your revised manuscript by Apr 11 2020 11:59PM. To enhance the reproducibility of your results, we recommend that if applicable you deposit your laboratory protocols in protocols.io, where a protocol can be assigned its own identifier (DOI) such that it can be cited independently in the future. For instructions see: http://journals.plos.org/plosone/s/submission-guidelines#loc-laboratory-protocols

We look forward to receiving your revised manuscript.

Kind regards,

Maret G Traber, PhD

Academic Editor

PLOS ONE

Additional Editor Comments (if provided):

Your manuscript describes an interesting approach to the continuing problem of fat malabsorption in patients with cystic fibrosis. The three reviewers were positive in their comments, but asked for some clarifications. One asked for a new statistical approach. Please address these comments and either justify your approach, or revise your statistical analysis.

Journal Requirements:

2. Thank you for including your ethics statement: This study was approved by the Children's Hospital of Philadelphia Institutional Review Board and each CF center (#4611). Verbal assent was obtained from subjects 6 to <18 years and written consent from parents/legal guardians of subjects <18 years.

3. Please provide additional details regarding participant consent. In the ethics statement in the Methods and online submission information, please ensure that you have specified whether written consent was informed (line 25).

4. Please add sub-sections to each part of your Methods section to aid in the reading of the manuscript.

"Supported by NIDDK (R44DK060302), and the Nutrition Center at the Children’s Hospital of Philadelphia. The project described was supported by the National Center for Research Resources, Grant UL1RR024134, and is now at the National Center for Advancing Translational Sciences, Grant UL1TR000003."

Please remove any funding-related text from the manuscript and let us know how you would like to update your Funding Statement. Currently, your Funding Statement reads as follows: none.

7. PLOS requires an ORCID iD for the corresponding author in Editorial Manager on papers submitted after December 6th, 2016. Please ensure that you have an ORCID iD and that it is validated in Editorial Manager. To do this, go to ‘Update my Information’ (in the upper left-hand corner of the main menu), and click on the Fetch/Validate link next to the ORCID field. This will take you to the ORCID site and allow you to create a new iD or authenticate a pre-existing iD in Editorial Manager. Please see the following video for instructions on linking an ORCID iD to your Editorial Manager account: https://www.youtube.com/watch?v=_xcclfuvtxQ

8. Please include a separate caption for each figure in your manuscript.

Reviewers' comments:

Reviewer's Responses to Questions

**Comments to the Author**

1. Is the manuscript technically sound, and do the data support the conclusions?

Reviewer #1: Yes

Reviewer #2: Partly

Reviewer #3: Yes

2. Has the statistical analysis been performed appropriately and rigorously? 

Reviewer #1: I Don't Know

Reviewer #2: Yes

Reviewer #3: Yes

3. Have the authors made all data underlying the findings in their manuscript fully available?

Reviewer #1: Yes

Reviewer #2: Yes

Reviewer #3: Yes

4. Is the manuscript presented in an intelligible fashion and written in standard English?

Reviewer #1: Yes

Reviewer #2: Yes

Reviewer #3: Yes

5. Review Comments to the Author

Reviewer #1: An team of nutrition investigators conducted a 2007-2011 double-blind placebo controlled dietary supplement study using LYM-X-SORBTM (LXS), a readily absorbable structured lipid technology developed to increase dietary fat absorption, in CF subjects with pancreatic insufficiency ages 5-18. The overall 12 month study demonstrated that the oral LXS supplementation was safe and more effective than the energy and micronutritent-matched placebo supplement and resulted in increase daily fat absorption and fasting plasma fatty acid status and growth as published in ref 9, related parts of the study being published in refs 8, 10 &11. In the present report the authors have harvested data gathered at baseline and the three month visit from 40 LXS supplemented and 46 placebo supplemented subjects. Findings in this secondary analysis showed LXS to be most effective in subjects with more severe baseline malabsorption of fat. The authors present solid data supporting their conclusions, the findings are of translational value to clinical medicine and the discussion is both informative and relevant. A story case is made for more widespread use of LXS in CF patients with continuing fat malabsorption after optimizations for dosing of PERT.

Comments

1. A major issue needing to be addressed by the authors concerns the modifying effects of CFTR modulator therapies on fat malabsorption. Unlike the subjects of a decade ago, most CF patients are now receiving CFTR corrector and potentiator therapies known to be modifiers of intestinal absorption... and this significantly impacts the translational value of this paper. There are now publications addressing this issue.

2. It would be useful if the authors could mention the effects of LXS to increase 20:5 and 22:6 absorption. To what extent are these values low in CF because of absorption issues vs. deficiencies in desaturases?

3. Line 127 and 180, what is CP?

4. Do the authors have any information on the effects of LXS on carotenoid absorption?

Reviewer #2: Stallings et al. performed a secondary analysis of a multicenter RCT focusing on the effects of 3-month supplementation with a readily absorbable structured lipid versus placebo in 66 patients with cystic fibrosis aged 10.3±3.0 years, which has been reported previously to be well tolerated and exert beneficial effects on fat absorption, growth and nutritional and essential fatty acid status.

In the present manuscript the effects in patients with coefficient of fat absorption (CFA) below versus above 88%, the median baseline CFA of the study group, were analyzed. Patients with lower than median CFA receiving the supplement showed improved CFA, anthropometry (weight, BMI z scores) and plasma essential fatty acids (linoleic, α-linolenic acid) and those receiving placebo only showed improved weight z scores (due to additional energy intake from the supplement), while patients with higher than median CFA (both treatment and placebo groups) had improvements in weight z scores and some plasma fatty acids.

The authors concluded that the supplement provided greater benefits to CF patients with more severe baseline residual dietary fat malabsorption and that it enhanced dietary energy absorption, weight gain and fatty acid status.

Overall, the RCT has been well designed and conducted using state-of-the-art methodologies and complying with human intervention study standards. However, this study has a few strong limitations:

(1) The main difficulty arises from the fact that the RCT was not designed and powered for detecting a difference in the treatment effects between patients with CFA below versus above the median CFA of the entire study group.

(2) The threshold of CFA 88% is not clinically or pathophysiology based, but determined by group characteristics and data distribution.

(3) There is an unbalanced distribution of 63% of the treatment group showing CFA <88%, while only 39% did so in the placebo group.

(4) Improvement of fat absorption at the level of the mean changes in CFA (7.5 � 7.2 %) and stool fat excretion (5.7 �7.2 g/d) reported for the group with CFA <88% receiving the supplement are clinically relevant, but the standard deviations are very high, suggesting that only a few patients may have responded that well.

(5) In the absence of dietary fatty acid intake data changes in plasma fatty acid concentrations are difficult to explain by improved intestinal fatty acid absorption.

Given that fat malabsorption-associated improvements are clinically relevant in CF patients, it is suggested that the authors address the following recommendations for further exploration of the results:

As a consequence of (1) some real differences might not have been detected while others could have been observed by chance. For instance, there were significant changes in plasma linoleic and alpha-linolenic acid in the CFA>88% group receiving placebo, which is difficult to explain. Student’s paired tests were applied for testing time effects within treatment groups. When using for instance 2-way repeated measures ANOVA across the treatment groups, the likelihood to observe significant changes by chance could be minimized.

To overcome (2) the limitation of an arbitrary threshold rather than one supported by clinical or pathophysiological criteria, it is suggested that differences in the treatment effects are studied across the entire range of baseline CFA instead of splitting the group by the median CFA and/or as an additional approach. It is suggested that the authors visualize and analyze the data in that way. This could possibly even help detect a real clinically relevant threshold (if such exists in this study).

Regarding (3), even though the differences in baseline CFA were not significant between treatment and placebo group, the impact of this imbalance on the results should be taken care of by additional statistical analysis.

Given (4) the high standard deviations of 7.5÷7.2 % for CFA and 5.7÷7.2 g/d for stool fat excretion in the CFA <88% and treatment group, a closer look at the data of this subgroup is suggested, including, for instance, the impact of absence versus presence of CF-associated cholestatic liver disease – which may have been more (or even exclusively) prevalent in CFA<88%.

(5) Even though the 3-day weighed dietary records performed before and after the intervention may be too short in duration to definitely exclude possible effects of changes in dietary intake, they could be helpful in this regard. It is therefore suggested to include fatty acid intake data in the present manuscript. Studying the improvements in plasma FA in relation to improvements in CFA by using regression analysis could help support a mechanistic explanation of improved intestinal absorption proposed in the manuscript.

Minor point:

λ-linolenic acid should read γ-linolenic acid.

Reviewer #3: This article is certainly very interesting and useful as it proposes to investigate whether the effect of 3 months of LXS treatment varied by the participants degree of dietary fat malabsorption at baseline. The paper is well written and highly readable, however they some minor comments worth addressing.

Minor

Perhaps not essential but it would it be good to include a rationale for looking at 3 month time-point. Clinical reason?, or more to do with immediate effects of the intervention?

It would perhaps be more clear and transparent that this was a subgroup analysis. Calling it a secondary analysis might imply that the analysis is based on one of the secondary objectives from the main RCT, but it isn’t.

In the methods section, it would be help if the sections were labelled clearly or made distinguishable, ‘outcomes’, ‘statistical analysis’

Result section, it’s worth mentioning that 86 participants completed 3 months (according to the Fig 1), however 66 children had data available for the ‘subgroup’ analysis.

Lines 118 – 121 (page 9), also give actual numbers as you have done for low CFA group.

6. PLOS authors have the option to publish the peer review history of their article (what does this mean?). If published, this will include your full peer review and any attached files.

Reviewer #1: No

Reviewer #2: No

Reviewer #3: No

---

## [Author Response · Author response to Decision Letter 0]

26 Mar 2020

Academic Editor

PLOS ONE

Additional Editor Comments (if provided):

Your manuscript describes an interesting approach to the continuing problem of fat malabsorption in patients with cystic fibrosis. The three reviewers were positive in their comments, but asked for some clarifications. One asked for a new statistical approach. Please address these comments and either justify your approach, or revise your statistical analysis.

Journal Requirements:

2. Thank you for including your ethics statement: This study was approved by the Children's Hospital of Philadelphia Institutional Review Board and each CF center (#4611). Verbal assent was obtained from subjects 6 to <18 years and written consent from parents/legal guardians of subjects <18 years.

We have added the full name of each of the other sites IRB:

All study visits were conducted at Children’s Hospital of Philadelphia (CHOP) at baseline, three, and 12 months and the protocol was approved by the CHOP Institutional Review Board (IRB) and each participating CF Center (Eastern Virginia Medical School IRB, Children’s National Medical Center IRB, Yale University School of Medicine IRB, University of Virginia IRB, Schneider Children’s Hospital IRB (now: Cohen Children’s Medical Center), St. Joseph’s Children’s Hospital IRB, and Monmouth Medical Center IRB).

We have added this information to the submission form.

3. Please provide additional details regarding participant consent. In the ethics statement in the Methods and online submission information, please ensure that you have specified whether written consent was informed (line 25).

We have amended the sentence to describe how consent was obtained: “Informed, verbal assent was obtained from subjects 6.0 to <18.0 years and informed, written consent from parents/legal guardians of subjects <18 years.”

4. Please add sub-sections to each part of your Methods section to aid in the reading of the manuscript.

We have added the following sub-sections: Participants, Inclusion/Exclusion Criteria, Design, Outcomes, and Statistics 

"Supported by NIDDK (R44DK060302), and the Nutrition Center at the Children’s Hospital of Philadelphia. The project described was supported by the National Center for Research Resources, Grant UL1RR024134, and is now at the National Center for Advancing Translational Sciences, Grant UL1TR000003."

Please remove any funding-related text from the manuscript and let us know how you would like to update your Funding Statement. Currently, your Funding Statement reads as follows: none.

 We have updated the funding statement in the online submission form and removed the funding information from the Acknowledgements section.

 We have updated the Competing Interests by adding this information after the “Acknowledgements” section in the manuscript as there was not a “Competing Interests” section within the online submission form.

7. PLOS requires an ORCID iD for the corresponding author in Editorial Manager on papers submitted after December 6th, 2016. Please ensure that you have an ORCID iD and that it is validated in Editorial Manager. To do this, go to ‘Update my Information’ (in the upper left-hand corner of the main menu), and click on the Fetch/Validate link next to the ORCID field. This will take you to the ORCID site and allow you to create a new iD or authenticate a pre-existing iD in Editorial Manager. Please see the following video for instructions on linking an ORCID iD to your Editorial Manager account: https://www.youtube.com/watch?v=_xcclfuvtxQ

 This has been added (https://orcid.org/0000-0002-2045-3002)

8. Please include a separate caption for each figure in your manuscript.

A caption has been added to the Figure 1 powerpoint file.

Reviewers' comments:

Reviewer's Responses to Questions

Comments to the Author

1. Is the manuscript technically sound, and do the data support the conclusions?

Reviewer #1: Yes

Reviewer #2: Partly

Reviewer #3: Yes

2. Has the statistical analysis been performed appropriately and rigorously? 

Reviewer #1: I Don't Know

Reviewer #2: Yes

Reviewer #3: Yes

3. Have the authors made all data underlying the findings in their manuscript fully available?

Reviewer #1: Yes

Reviewer #2: Yes

Reviewer #3: Yes

4. Is the manuscript presented in an intelligible fashion and written in standard English?

Reviewer #1: Yes

Reviewer #2: Yes

Reviewer #3: Yes 

5. Review Comments to the Author

Reviewer #1: An team of nutrition investigators conducted a 2007-2011 double-blind placebo controlled dietary supplement study using LYM-X-SORBTM (LXS), a readily absorbable structured lipid technology developed to increase dietary fat absorption, in CF subjects with pancreatic insufficiency ages 5-18. The overall 12 month study demonstrated that the oral LXS supplementation was safe and more effective than the energy and micronutritent-matched placebo supplement and resulted in increase daily fat absorption and fasting plasma fatty acid status and growth as published in ref 9, related parts of the study being published in refs 8, 10 &11. In the present report the authors have harvested data gathered at baseline and the three month visit from 40 LXS supplemented and 46 placebo supplemented subjects. Findings in this secondary analysis showed LXS to be most effective in subjects with more severe baseline malabsorption of fat. The authors present solid data supporting their conclusions, the findings are of translational value to clinical medicine and the discussion is both informative and relevant. A story case is made for more widespread use of LXS in CF patients with continuing fat malabsorption after optimizations for dosing of PERT.

Comments

1. A major issue needing to be addressed by the authors concerns the modifying effects of CFTR modulator therapies on fat malabsorption. Unlike the subjects of a decade ago, most CF patients are now receiving CFTR corrector and potentiator therapies known to be modifiers of intestinal absorption... and this significantly impacts the translational value of this paper. There are now publications addressing this issue.

This is a great point. We conducted a literature search of modulator therapies and fat malabsorption and there is little information on the impact of fat absorption. A recent study from our group (Stallings VA et al., J Peds, 2018) that evaluated CFA and fecal calprotectin (marker of inflammation) in subjects with gating mutations and showed that CFA increased significantly (+1.5%) in participants who were pancreatic insufficient (PI) after ivacaftor treatment, and calprotectin was reduced in both PI and PS. Studies also report that Ivacaftor can alter microbial communities (Ooi CY et al., Sci Rep, 2018) and partially restored this disruption of bile acid homeostasis (van de Peppel IP et al., J Cyst Fibros, 2019), which could affect fat absorption. However, in individuals with more severe (non-gating) mutations, there are no data related to fat malabsorption. There is evidence of improved intestinal pH (Gelfond D et al., Clin Transl Gastroenterol, 2017) which could help pancreatic enzyme function, but this is speculative.

A comment about this this issue of impact of modulator therapy was added to the introduction and discussion.

2. It would be useful if the authors could mention the effects of LXS to increase 20:5 and 22:6 absorption. To what extent are these values low in CF because of absorption issues vs. deficiencies in desaturases?

Thank you for this suggestion. We added plasma eicosapentaenoic acid (EPA: C20:5w3), docosahexaenoic acid (DHA: C22:6w3), and docosapentaenoic acid (DPA: C22:5w3) concentrations at baseline comparing both CFA groups and treatment (randomization) groups in Table 1, and also the baseline, 3 month, and 3-month changes for each treatment/CFA group in Table 2.

There is evidence that fatty acid desaturase activity is actually increased in people with cystic fibrosis via stimulation of the AMP-activated protein kinase in the absence of a functional CFTR protein (Freedman SD et al., Proc Natl Acad Sci U S A. 1999; Mimoun M et al., J Nutr, 2009; Njoroge SW et al., Biochim Biophys Acta, 2011; Seegmiller AC, Int J Mol Sci, 2014). Low values in CF are a result of poor absorption and the impact of altered (increased and decreased pathways) metabolism may play a role.

3. Line 127 and 180, what is CP?

Thank you for this question. We have corrected this error to “CF”, not “CP”.

4. Do the authors have any information on the effects of LXS on carotenoid absorption?

Upon review, we do not have data on the effects of LXS on carotenoid absorption.

Reviewer #2: Stallings et al. performed a secondary analysis of a multicenter RCT focusing on the effects of 3-month supplementation with a readily absorbable structured lipid versus placebo in 66 patients with cystic fibrosis aged 10.3±3.0 years, which has been reported previously to be well tolerated and exert beneficial effects on fat absorption, growth and nutritional and essential fatty acid status.

In the present manuscript the effects in patients with coefficient of fat absorption (CFA) below versus above 88%, the median baseline CFA of the study group, were analyzed. Patients with lower than median CFA receiving the supplement showed improved CFA, anthropometry (weight, BMI z scores) and plasma essential fatty acids (linoleic, α-linolenic acid) and those receiving placebo only showed improved weight z scores (due to additional energy intake from the supplement), while patients with higher than median CFA (both treatment and placebo groups) had improvements in weight z scores and some plasma fatty acids.

The authors concluded that the supplement provided greater benefits to CF patients with more severe baseline residual dietary fat malabsorption and that it enhanced dietary energy absorption, weight gain and fatty acid status.

Overall, the RCT has been well designed and conducted using state-of-the-art methodologies and complying with human intervention study standards. However, this study has a few strong limitations:

(1) The main difficulty arises from the fact that the RCT was not designed and powered for detecting a difference in the treatment effects between patients with CFA below versus above the median CFA of the entire study group.

We agree that the RCT a priori plan did not specify this analysis. However, the RCT results and interest in applications from our clinical colleagues led us to conduct this post hoc subgroup exploratory analysis. These facts are presented in the methods section, so the reader is aware that this was an exploratory analysis with associated limitations. 

(2) The threshold of CFA 88% is not clinically or pathophysiology based, but determined by group characteristics and data distribution.

We agree there is no evidence-based, generally accepted cut-point for ‘typical’ or ‘best attainable’ CFA in people CF and PI. In addition, a cut-point is unlikely to be established since the CFA test is employed almost exclusively for research studies and for FDA evaluations of new pancreatic enzyme efficacy (all studies with small sample sizes). CFA reference value for healthy individuals is generally accepted as 93% or greater. A soon to be published paper from our group (Bashaw, et al 2020), confirms this value is still meaningful in a contemporary sample of healthy volunteers using the same study method as in this report. 

After discussions with our biostatistician and consideration of feasibility (sample size, contrast between groups) with our data set, the approach to use the median for this subgroup analysis was employed.

The following comments were added to the limitations section: 

The low and high CFA groups in this study were defined for this study as the median CFA at baseline (88%) to provide a balanced distribution of participants. There is no generally accepted or evidence based definition for classification of individuals with CF and PI at high and low risk of fat malabsorption by CFA value. CFA is used mostly in the research setting and for FDA evaluation of new pancreatic enzyme products.

(3) There is an unbalanced distribution of 63% of the treatment group showing CFA <88%, while only 39% did so in the placebo group.

Agree, and this is related to the original (Encala/placebo group) randomization procedure for all subjects with sex/age blocks. It did not include CFA, as CFA was an outcome variable. We have included this in the limitations section.

(4) Improvement of fat absorption at the level of the mean changes in CFA (7.5 � 7.2 %) and stool fat excretion (5.7 �7.2 g/d) reported for the group with CFA <88% receiving the supplement are clinically relevant, but the standard deviations are very high, suggesting that only a few patients may have responded that well.

A higher standard deviation in the low CFA group was expected given that the range was wider (52-87%) than that of the high CFA where the range was lower (>88 and <100%) and has a ceiling effect since no one is above 100%. As part of preparing this response, we carefully reviewed the CFA for each individual; there were no outliers nor implausible data. In subjects who improved, all but 2 subjects improved between +1 and +16%, except 2 of 31 subjects, who improved >16%. In subjects who had decreased CFA, all subjects decreased between -1 and -16%, except 2 of 31 subjects who decreased >-16%. We confirmed that the results were not driven by ‘only a few patients may have responded that well’.

(5) In the absence of dietary fatty acid intake data changes in plasma fatty acid concentrations are difficult to explain by improved intestinal fatty acid absorption.

Given that fat malabsorption-associated improvements are clinically relevant in CF patients, it is suggested that the authors address the following recommendations for further exploration of the results:

As a consequence of (1) some real differences might not have been detected while others could have been observed by chance. For instance, there were significant changes in plasma linoleic and alpha-linolenic acid in the CFA>88% group receiving placebo, which is difficult to explain. Student’s paired tests were applied for testing time effects within treatment groups. When using for instance 2-way repeated measures ANOVA across the treatment groups, the likelihood to observe significant changes by chance could be minimized.

We appreciate the comment to reconsider the analysis. We are evaluating changes over time and this analysis is only a two measurement/data set-- from baseline to 3 months. A repeated measure ANOVA would be most suitable when there are comparisons among more than two groups or three or more time points (Park E et al., Korean J Lab Med, 2009; Bergh DD et al., Acad Manage J, 1995). We have chosen Student’s paired t-test for the time effects within each treatment groups (change from baseline to 3-months), and the Student’s unpaired t test for the difference between treatment groups in the 3-month change as the most suitable analysis since only two groups are compared. Therefore, we consider our analysis appropriate for this data set and subgroup analysis. 

To overcome (2) the limitation of an arbitrary threshold rather than one supported by clinical or pathophysiological criteria, it is suggested that differences in the treatment effects are studied across the entire range of baseline CFA instead of splitting the group by the median CFA and/or as an additional approach. It is suggested that the authors visualize and analyze the data in that way. This could possibly even help detect a real clinically relevant threshold (if such exists in this study).

Baseline CFA Range Question: As a standard component of our analysis, we did visualize the baseline CFA data, looking for outliers, implausible data, and as you suggested, potentially obvious cut-points to consider. The results showed: n=4 subjects between 50-50.9% (lowest was 52%); n=5 between 60-60.9%; n=11 between 70-70.9%; and n=21 between 80-90%. There were n = 25 >90%, including n=14 between 90-<93%, and n=11 in the reference range for healthy people (≥93%). Our assessment was there was a good distribution across the %CFA results, there was no suggestion of specific clustering that was not expected in this outcome in CF/PI subjects, and all data were used in the analysis. The approach to use the median, with balanced sample size in both groups, was selected for the subgroup/exploratory analysis to determine if there was a different response to Encala vs Placebo intervention in subjects with more severe fat malabsorption at baseline.

We established a working definition of low and high baseline CFA for ‘use in this project’, and do not suggest it be used as a clinical endpoint. In the methods section, we have made it clear, that the cut point/definition is for the purpose of data analysis in the paper. CFA is not used in clinical care (burden of complete 3-day stool and accurate diet intake), so it is unlikely that clinicians will misinterpret/misuse this information for clinical care. 

Regarding (3), even though the differences in baseline CFA were not significant between treatment and placebo group, the impact of this imbalance on the results should be taken care of by additional statistical analysis.

We recognize the unbalanced groups, however, this is an exploratory sub-group analysis of a study with balanced groups initially (110 subjects randomized to treatment [n=54] and placebo [n=56]). We suggest this analysis, with the reviewers-directed clarification and additions to the limitation section, is an acceptable approach for secondary, subgroup analysis. We agree and recommend future studies consider similar questions with appropriate design and adequate sample size.

Given (4) the high standard deviations of 7.5÷7.2 % for CFA and 5.7÷7.2 g/d for stool fat excretion in the CFA <88% and treatment group, a closer look at the data of this subgroup is suggested, including, for instance, the impact of absence versus presence of CF-associated cholestatic liver disease – which may have been more (or even exclusively) prevalent in CFA<88%.

Please see the response above (4) regarding the standard deviation values mentioned or we’ve also copied it here:

A higher standard deviation in the lower CFA group was expected given that the range was wider (52-87%) than that of the high CFA where the range was lower (>88 and <100%) and has a ceiling effect since no one is above 100%. As part of preparing this response, we carefully reviewed the CFA for each individual; there were no outliers nor implausible data. In subjects who improved, all but 2 subjects improved between +1 and +16%, except 2 of 31 subjects, who improved >16%. In subjects who had decreased CFA, all subjects decreased between -1 and -16%, except 2 of 31 subjects who decreased >-16%. 

Low CFA Group: Upon review of individual, raw data, there were no outliers. Rather, there is a large range of CFAs in the low CFA group (52-87%). 

Significant liver disease and any other significant diagnosis that might impact dietary intake, growth or body composition were exclusion criteria for subjects to participate in this study. There were no subjects with cholestatic liver disease.

(5) Even though the 3-day weighed dietary records performed before and after the intervention may be too short to definitely exclude possible effects of changes in dietary intake, they could be helpful in this regard. It is therefore suggested to include fatty acid intake data in the present manuscript. Studying the improvements in plasma FA in relation to improvements in CFA by using regression analysis could help support a mechanistic explanation of improved intestinal absorption proposed in the manuscript.

In nutrition studies, the prospective, three day, weighted food record is considered a high quality method to provide the food intake evidence/data (Johnson RK, Obesity, 2012;Yang JH et al., Nutr Res Pract. 2010; Crawford PB et al., JAND, 1994). Also note, our study specific research staff trains the subjects/families on food intake procedure, provides the digital food scales, measuring cups and spoons, and conducts the follow-up call when indicated, to clarify uncertain entry. The protocol did not call for modification to usual dietary intake, and the interventions (Encala and placebo products) were designed to be added to subjects self-selected, routine/preferred foods. 

As suggested, we added the total, saturated, monounsaturated and polyunsaturated dietary fat intake data to Table 2 and dietary data from the full cohort of participants was reported in Stallings VA et al., J Pediatr Gastroenterol Nutr, 2016, supplemental Table 1. There were no significant differences in intake between groups for total, monounsaturated fat, polyunsaturated fat, or saturated fat classes. The subgroup subjects here are representative of the larger population and therefore, the lack of difference in dietary is not unexpected. Therefore, we respectfully disagree that changes in dietary intake were different between groups, and so unlikely to be the etiology of the plasma results.

Minor point:

λ-linolenic acid should read γ-linolenic acid.

Thank you. We have corrected this.

Reviewer #3: This article is certainly very interesting and useful as it proposes to investigate whether the effect of 3 months of LXS treatment varied by the participants degree of dietary fat malabsorption at baseline. The paper is well written and highly readable, however they some minor comments worth addressing.

Minor

Perhaps not essential but it would it be good to include a rationale for looking at 3 month time-point. Clinical reason?, or more to do with immediate effects of the intervention?

The three-month time-point was selected because it was sufficient time for the intervention to have effect on the outcomes of interest, and the 3 month protocol visit provided the a sufficient number of subjects with all the data elements needed for the subgroup analysis. 

It would perhaps be more clear and transparent that this was a subgroup analysis. Calling it a secondary analysis might imply that the analysis is based on one of the secondary objectives from the main RCT, but it isn’t.

We agree. We have changed the language from “secondary” to “subgroup”.

In the methods section, it would be help if the sections were labelled clearly or made distinguishable, ‘outcomes’, ‘statistical analysis’

As suggested, we have created the following labels: Participants, Inclusion/Exclusion Criteria, Design, Outcomes, and Statistics.

Result section, it’s worth mentioning that 86 participants completed 3 months (according to the Fig 1), however 66 children had data available for the ‘subgroup’ analysis.

We have included the following sentence to the results:

Eighty-six participants completed three months on either Encala or placebo, but this subgroup analysis included 66 children and adolescents (10.5±3.0 yrs, 39% female) with both baseline and 3-month visit CFA assessments (stool and diet records). 

Lines 118 – 121 (page 9), also give actual numbers as you have done for low CFA group.

We added the actual values and amended the low CFA group language to:

For subjects in the higher baseline CFA subgroup, CFA% (-1.2±4.8% on placebo and -4.4±6.3% on Encala), stool fat (0.3±4.5 g/d on placebo and 2.9±7.3 g/d on Encala) and dietary fat intake did not change with either placebo or Encala treatment. WAZ improved in both groups (0.13±0.28 for placebo and 0.14±0.21 for Encala, p<0.05).

---

## [Decision Letter · Decision Letter 1]

9 Apr 2020

PONE-D-19-35347R1

Improved Residual Fat Malabsorption and Growth in Children with Cystic Fibrosis Treated with a Novel Oral Structured Lipid Supplement: A Randomized Controlled Trial

PLOS ONE

Dear Dr. Tindall,

Thank you for submitting your manuscript to PLOS ONE. After careful consideration, we feel that it has merit but does not fully meet PLOS ONE’s publication criteria as it currently stands. Therefore, we invite you to submit a revised version of the manuscript that addresses the points raised during the review process.

We would appreciate receiving your revised manuscript by May 24 2020 11:59PM. To enhance the reproducibility of your results, we recommend that if applicable you deposit your laboratory protocols in protocols.io, where a protocol can be assigned its own identifier (DOI) such that it can be cited independently in the future. For instructions see: http://journals.plos.org/plosone/s/submission-guidelines#loc-laboratory-protocols

We look forward to receiving your revised manuscript.

Kind regards,

Maret G Traber, PhD

Academic Editor

PLOS ONE

Additional Editor Comments (if provided):

Please address the outstanding concerns from one of the reviewers, or provide a rationale why you think this action is unnecessary.

Reviewers' comments:

Reviewer's Responses to Questions

**Comments to the Author**

1. If the authors have adequately addressed your comments raised in a previous round of review and you feel that this manuscript is now acceptable for publication, you may indicate that here to bypass the “Comments to the Author” section, enter your conflict of interest statement in the “Confidential to Editor” section, and submit your "Accept" recommendation.

Reviewer #2: (No Response)

2. Is the manuscript technically sound, and do the data support the conclusions?

Reviewer #2: Yes

3. Has the statistical analysis been performed appropriately and rigorously? 

Reviewer #2: Yes

4. Have the authors made all data underlying the findings in their manuscript fully available?

Reviewer #2: Yes

5. Is the manuscript presented in an intelligible fashion and written in standard English?

Reviewer #2: Yes

6. Review Comments to the Author

Reviewer #2: The authors have addressed the concerns raised by this reviewer in their point-by-point reply and have added corresponding additions to their manuscript.

In contrast to the statement in the reply "As suggested, we added the total, saturated, monounsaturated and polyunsaturated dietary fat intake data to Table 2" and "There were no significant differences in intake between groups for total, monounsaturated fat, polyunsaturated fat, or saturated fat classes." these additional data do not appear in this table. These data should be added as described.

While it was proposed that "Regarding the unbalanced distribution of 63% of the treatment group showing CFA <88%, while only 39% did so in the placebo group ... this is related to the original (Encala/placebo group) randomization procedure for all subjects with sex/age blocks ... It did not include CFA, as CFA was an outcome variable ... We have included this in the limitations section", this explanation is still missing in the limitations section and should be appropriately addressed in a re-revised version of the manuscript.

Even though the inclusion/exclusion criteria have been previously reported (refs. 10-12), it would be helpful for the readers to include information regarding "Significant liver disease and any other significant diagnosis that might impact dietary intake, growth or body composition were exclusion criteria for subjects to participate in this study. There were no subjects with cholestatic liver disease" provided in the reply also in the manuscript as this is highly relevant for the present study.

7. PLOS authors have the option to publish the peer review history of their article (what does this mean?). If published, this will include your full peer review and any attached files.

Reviewer #2: No

---

## [Author Response · Author response to Decision Letter 1]

15 Apr 2020

Reviewer #2: The authors have addressed the concerns raised by this reviewer in their point-by-point reply and have added corresponding additions to their manuscript.

In contrast to the statement in the reply "As suggested, we added the total, saturated, monounsaturated and polyunsaturated dietary fat intake data to Table 2" and "There were no significant differences in intake between groups for total, monounsaturated fat, polyunsaturated fat, or saturated fat classes." these additional data do not appear in this table. These data should be added as described.

We have amended the table to include these dietary fats and the statement "There were no significant differences in intake between groups for total, monounsaturated fat, polyunsaturated fat, or saturated fat classes” remains accurate.

While it was proposed that "Regarding the unbalanced distribution of 63% of the treatment group showing CFA <88%, while only 39% did so in the placebo group ... this is related to the original (Encala/placebo group) randomization procedure for all subjects with sex/age blocks ... It did not include CFA, as CFA was an outcome variable ... We have included this in the limitations section", this explanation is still missing in the limitations section and should be appropriately addressed in a re-revised version of the manuscript.

We have added the following sentence to the discussion: “The unbalanced distribution was related to the original Encala™/placebo randomization procedure for all subjects with sex/age blocks that did not include CFA, as CFA was an outcome variable.” (line 227-229 in the marked up version)

Even though the inclusion/exclusion criteria have been previously reported (refs. 10-12), it would be helpful for the readers to include information regarding "Significant liver disease and any other significant diagnosis that might impact dietary intake, growth or body composition were exclusion criteria for subjects to participate in this study. There were no subjects with cholestatic liver disease" provided in the reply also in the manuscript as this is highly relevant for the present study.

 We have added the following sentence under “inclusion/exclusion criteria”: “Liver disease and any other significant diagnosis that might impact dietary intake, growth or body composition were exclusion criteria for subjects to participate in this study. There were no subjects with cholestatic liver disease.”

---

## [Editor Report · Decision Letter 2]

21 Apr 2020

Improved Residual Fat Malabsorption and Growth in Children with Cystic Fibrosis Treated with a Novel Oral Structured Lipid Supplement: A Randomized Controlled Trial

PONE-D-19-35347R2

Dear Dr. Stallings,

We are pleased to inform you that your manuscript has been judged scientifically suitable for publication and will be formally accepted for publication once it complies with all outstanding technical requirements.

With kind regards,

Maret G Traber, PhD

Academic Editor

PLOS ONE

Additional Editor Comments (optional):

The authors have responded acceptably.
---

## [Editor Report · Acceptance letter]

28 Apr 2020

PONE-D-19-35347R2 

Improved Residual Fat Malabsorption and Growth in Children with Cystic Fibrosis Treated with a Novel Oral Structured Lipid Supplement: A Randomized Controlled Trial 

Dear Dr. Stallings:

I am pleased to inform you that your manuscript has been deemed suitable for publication in PLOS ONE. Congratulations! Your manuscript is now with our production department. 

With kind regards,

on behalf of

Professor Maret G Traber 

Academic Editor

PLOS ONE